# Generalized Householder Transformations

**DOI:** 10.3390/e24030429

**Published:** 2022-03-19

**Authors:** Karl Svozil

**Affiliations:** Institute for Theoretical Physics, TU Wien, Wiedner Hauptstrasse 8-10/136, 1040 Vienna, Austria; svozil@tuwien.ac.at

**Keywords:** Householder transformation, expectation value, probability distribution, affine transformation

## Abstract

The Householder transformation, allowing a rewrite of probabilities into expectations of dichotomic observables, is generalized in terms of its spectral decomposition. The dichotomy is modulated by allowing more than one negative eigenvalue or by abandoning binaries altogether, yielding generalized operator-valued arguments for contextuality. We also discuss a form of contextuality by the variation of the functional relations of the operators, in particular by additivity.

## 1. From probabilities to expectations

A standard way to recast classical probabilities $p\in \left[0,1\right]$ into expectations $E\in \left[-a,a\right]$ of two-valued—indeed, $\left\{-a,a\right\}$-valued, observables—is in terms of affine transformations $E_{a}\left(p\right)=a\left(2p-1\right)$, amounting to a doubling of the probability and a shift by minus one, times *a*. (Often the physical units in terms of which observables are measured are chosen to be such that a=1.) This can be motivated by the linearity of classical probabilities, which can be defined as the convex polytope of “extreme cases” or truth assignments, symbolized by two-valued measures $v\in \left\{0,1\right\}$.

It is an interesting property of quantum mechanics that the dimensionality n∈N of the associated Hilbert space $C^{n}$ is determined by the finest resolution of its contexts or “maximal observables”: a context contains an exhaustive (also known as maximal or complete) set of mutually exclusive elementary observables. Each one of these elementary observables is identifiable by an elementary proposition, which in turn is formalizable by a one-dimensional orthogonal projection operator 𝗙 that is both self-adjoint, as well as idempotent, that is $𝗙={𝗙}^{†}$, where † represents the Hermitian adjoint (also known as conjugate), and ${𝗙}^{2}=𝗙$, respectively. Thereby, n=2 associated with dichotomic observables just represents a bound from below for nontrivial predictions. However, for n>2, there are no preferred Leibnizian “dyadic” schemes, such as bases, to represent and encode vectors or pure states in *n*-dimensional Hilbert spaces: neither the dimensionality—suggesting rather an *n*-ary encoding—nor the scalar product (nor completeness) yield any such preference, albeit that arbitrary rotations (unitary transformations) in *n* dimensions can be obtained (and parameterized [1]) by the serial composition of rotations (unitary transformations) in two-dimensional subspaces of $C^{n}$.

Therefore, it might not be too far-fetched to ask which constructions might provide generalizations of the aforementioned affine transformations in arbitrary dimensions. In particular, what presents, at least to some degree of semblance, the quantum mechanical counterparts of classical expectations from probabilities mentioned earlier?

An answer can be given in terms of the so-called Householder transformations (e.g., [2]) as follows. The respective techniques are well developed, but may be less known in the quantum foundations community, so a review at the beginning seems in order. We shall then proceed to modifications of Householder transformations to nondichotomous, multiple eigenvalues.

Let $|x\rangle \in C^{n}$ be a nonzero vector and ${𝗙}_{x}={\left(\langle x|x\rangle \right)}^{-1}\left|x\rangle \langle x\right|$ the respective orthogonal projection operator. The Householder transformation ${𝗨}_{x}$ is defined by:(1)$${𝗨}_{x}=𝟙-2{𝗙}_{x}=𝟙-2{\left(\langle x|x\rangle \right)}^{-1}|x\rangle \langle x|.$$

If |x〉 is a unit vector, then ${𝗨}_{x}=𝟙-2|x\rangle \langle x|$.

The following properties can be asserted by direct proofs:(i)${𝗨}_{x}$ is Hermitian; that is, ${𝗨}_{x}={𝗨}_{x}^{†}$;(ii)${𝗨}_{x}$ is unitary, that is,
(2)$$\begin{array}{c} {𝗨}_{x}{𝗨}_{x}^{†}={𝗨}_{x}^{†}{𝗨}_{x}={𝗨}_{x}{𝗨}_{x}\\ =\left(𝟙-2{\left(\langle x|x\rangle \right)}^{-1}\left|x\rangle \langle x\right|\right)\left(𝟙-2{\left(\langle x|x\rangle \right)}^{-1}\left|x\rangle \langle x\right|\right)\\ =𝟙-4{\left(\langle x|x\rangle \right)}^{-1}\left|x\rangle \langle x\right|+4{\left(\langle x|x\rangle \right)}^{-1}\left|x\rangle \langle x\right|=𝟙; \end{array}$$(iii)Hence, ${𝗨}_{x}$ is involutory: ${𝗨}_{x}^{-1}={𝗨}_{x}$;(iv)The eigensystem of ${𝗨}_{x}$ has two eigenvalues ±1:−1:For the eigenvector |x〉 of ${𝗨}_{x}$, with ${𝗨}_{x}|x\rangle =\left(𝟙-2{\left(\langle x|x\rangle \right)}^{-1}\left|x\rangle \langle x\right|\right)|x\rangle =|x\rangle -2|x\rangle =-|x\rangle$, the associated eigenvalue is −1;+1:The remaining n−1 mutually orthogonal eigenvectors span the n−1-dimensional subspace orthogonal to |x〉. Every vector in that subspace has eigenvalue +1. (For n>2, the spectrum is degenerate.)Stated differently, for all vectors orthogonal to |x〉, the Householder transformation ${𝗨}_{x}$ acts as the identity; for |x〉, the Householder transformation ${𝗨}_{x}$ acts as a reflection on the one-dimensional subspace spanned by |x〉;(v)Since the determinant of a matrix is the product of its eigenvalues, the determinant of a Householder transformation is −1;(vi)If $C=\{|e_{1}\rangle ,|e_{2}\rangle ,\ldots ,|e_{n}\rangle \}$ is an orthonormal basis formalizing a context, then the succession of the respective Householder transformations renders negative unity, that is,
(3)$$\begin{array}{c} {𝗨}_{e_{1}}{𝗨}_{e_{2}}\cdots {𝗨}_{e_{n}}=\left(𝟙-2|e_{1}\rangle \langle e_{1}|\right)\left(𝟙-2|e_{2}\rangle \langle e_{2}|\right)\cdots \left(𝟙-2|e_{n}\rangle \langle e_{n}|\right)\\ =𝟙-2\underset{𝟙}{\underbrace{\left(|e_{1}\rangle \langle e_{1}\left|+\right|e_{2}\rangle \langle e_{2}\left|+\cdots +\right|e_{n}\rangle \langle e_{n}|\right)}}=-𝟙. \end{array}$$

For the sake of an example, let $|z\rangle ={\left(\begin{array}{c} 1,1 \end{array}\right)}^{⊺}$, so that the corresponding Householder transformation can be written in matrix form as:$${𝗨}_{z}=𝟙-2{\left(\langle z|z\rangle \right)}^{-1}\left|z\rangle \langle z\right|\equiv \left(\begin{array}{cc} 1 & 0\\ 0 & 1 \end{array}\right)-2{\left(2\right)}^{-1}\left(\begin{array}{cc} 1 & 1\\ 1 & 1 \end{array}\right)=-\left(\begin{array}{cc} 0 & 1\\ 1 & 0 \end{array}\right).$$

Take $|x\rangle ={\left(\begin{array}{c} 2,1 \end{array}\right)}^{⊺}$, so that $|y\rangle =-{\left(\begin{array}{c} 1,2 \end{array}\right)}^{⊺}$: this “reflected” vector |y〉 and the original vector |x〉 have the same length or norm. The component of |y〉 along |z〉 is reversed, whereas its component orthogonal to |z〉 remains the same. This situation is depicted in Figure 1.

Because of (iii), if |x〉≠|y〉 are two vectors in $R^{n}$ with identical length or norm ∥x∥=∥y∥, then there exists a remarkable “symmetry delivered by” a Householder transformation ${𝗨}_{z}$ such that ${𝗨}_{z}|x\rangle =|y\rangle$ and ${𝗨}_{z}{𝗨}_{z}|x\rangle ={𝗨}_{z}|y\rangle =|x\rangle$. For this to hold, the vector |z〉 needs to be a vector equal to |x〉−|y〉: $\left(𝟙-2{\left(\langle z|z\rangle \right)}^{-1}\left|z\rangle \langle z\right|\right)|x\rangle =|y\rangle$ and $|x\rangle =\left(𝟙-2{\left(\langle z|z\rangle \right)}^{-1}\left|z\rangle \langle z\right|\right)|y\rangle$, resulting in ${\left(\langle z|z\rangle \right)}^{-1}\left|z\rangle \langle z\right|\left(|x\rangle -|y\rangle \right)=|x\rangle -|y\rangle$, and thus, |z〉=|x〉−|y〉. (For |x〉=|y〉, identify with |z〉 a vector orthogonal to |x〉=|y〉.) This is not true for $C^{n}$, as for instance, there exists no |z〉 that would render ${𝗨}_{z}|x\rangle =i|x\rangle$ for nonzero |x〉, and an additional unitary transformation is required.

This gives rise to the orthonormalization of a set of *k* linear independent nonzero vectors $S=\{|s_{1}\rangle ,|s_{2}\rangle ,\ldots ,|s_{k}\rangle \}$ in $R^{n}$ by taking some orthonormal basis $C=\left\{e_{1},e_{2},\ldots ,e_{n}\right\}\equiv \{|e_{1}\rangle ,|e_{2}\rangle ,\ldots ,|e_{n}\rangle \}$, choosing *k* vectors thereof—say, the first *k* vectors of the standard Cartesian coordinate system—and identifying $|s_{i}\rangle$ with $|x_{i}\rangle$, and (the extra factor $∥s_{i}∥$ serves to make the vector of equal length or norm) $|y_{i}\rangle$ with $∥s_{i}∥|e_{i}\rangle$, thereby constructing a Householder transformation followed by normalization (through division by $∥s_{i}∥$) ${𝗨}_{z_{i}}$ of $|s_{i}\rangle \overset{{𝗨}_{z_{i}}}{\mapsto }|e_{i}\rangle$ with the respective $|z_{i}\rangle =|s_{i}\rangle -∥s_{i}∥|e_{i}\rangle$. This kind of orthonormalization may yield a span “outside” of the subspace spanned by the “original” vectors.

Cabello used the Householder transformation to argue for what he called “state-independent quantum contextuality” [3,4]. Thereby, in a first construction step, all $2^{16}$ possible classical value assignments of the elementary propositions $a_{1},\cdots ,a_{16}\in \left\{-1,1\right\}$ depicted in Figure 2, grouped into the nine contexts $C_{1},\ldots ,C_{9}$, are enumerated. In a second step, for each one of the nine contexts, the respective four (per context) possible classical value assignments of the elementary propositions are multiplied. In a third step, these nine (per classical value assignment) products are added together. As a result, each of the $2^{16}$ valuations yields a number, an integer between the algebraically maximal values −9 and 9—bounds obtained from the number of the (nine) contexts involved.

As it turns out, 9216 value assignments render the number −7, and none render −8 or −9. However, these classical value assignments are not admissible [5] in the sense of (iv) mentioned earlier—an ad hoc assumption—as there does not exist a classical (non-contextual) two-valued {0,1}-state on these 18 observables in nine contexts, which would allow a translation into a {−1,1}-value assignment such that each context contains exactly one element that is assigned the value “−1”, and all other elements of that context are assigned the value “+1”. For the sake of anecdotal demonstration (no proof), Figure 2 contains an “illegal” value assignment that renders the maximal value of seven of the sum of the products of all value assignments within the nine contexts.

Indeed, relative to admissibility, state-independent quantum contextuality is a corollary of the Kochen–Specker theorem for configurations without any two-valued states. Because in this case, no (homomorphic) translation from admissible two-valued {0,1}-states *p* into two-valued {−1,1}-observables *E* with affine E(p)=2p−1 exist.

In the relaxed case, admissibility can be violated—in particular, by an ad hoc breach of exclusivity, thereby allowing more than one value assignment “1” per context—while at the same time maintaining noncontextuality (at the intertwining observables). State-independent quantum contextuality can only be counterfactually postulated if and only if the quantum-Householder-transformation-based predictions—equal to the (modulus of the) number of contexts involved—are *not* realizable by classical noncontextual, admissible, or inadmissible value assignments. Therefore, the sum of all products of observables within all contexts should not reach its algebraic maximal obtainable value. (As noted earlier, this maximal obtainable value is just the number of contexts involved.) That implies that it should not be possible to require the number of noncontextual value assignments “−1” within each given context to be odd. As a result, strictly bi-connected (indeed, even-number-connected) Kochen–Specker configurations involving an odd number of contexts always exhibit state-independent quantum contextuality. The proof is similar to the indirect parity proof of the Kochen–Specker theorem for the configuration introduced by Cabello, Estebaranz, and García-Alcaine [6]: for a proof by contradiction, suppose the products of observables within all contexts are multiplied. On the one hand, since by assumption, there are odd contexts, each contributing a factor −1, this number—the odd product of products—should be −1. On the other hand, by bi- or even-connectivity, the product of products contains only squares or even multiples of factors, which return +1—a complete contradiction.

Figure 2 contains an instance of the classical inadmissible value assignment that cannot reach the algebraic maximal sum, as would be required by the quantum Householder transformation prediction. Further methods to obtain such configurations based on parity proofs were discussed by Waegell, Aravind, Megill, and Pavičić [7,8,9]. The Greenberger–Horne–Zeilinger operator theorem is based on a similar argument [10,11].

For all other multi-context configurations allowing two-valued states—even with a nonseparable or unital set of two-valued states—and the translation from {0,1}-states into two-valued {−1,1}-observables, there is no state-independent quantum contextuality. For other operator-valued assignments, see, for instance [4,12].

I shall leave open the question of how convincing and applicable to counterfactual arguments such inadmissible value assignments—even in their operator-valued translations—might be. At the moment, I am inclined to understand such situations and configurations rather in terms of the Kochen–Specker theorem [13], or quantitatively about the associated chromatic number, that is in terms of how many colors are needed to separate elements in the respective contexts [14].

A quantum realization of the Cabello, Estebaranz, and García-Alcaine [3,6] configuration is a faithful orthogonal representation [15,16,17] that includes 18 unit vectors or associated one-dimensional orthogonal projection operators ${𝗙}_{i}=|a_{i}\rangle \langle a_{i}|$, with 1≤i≤18 as vector labels of the hypergraph depicted in Figure 2, whereby the adjacency of hypergraph vertices is translated into the orthogonality of the vectors serving as their labels.

As we learned in (vi), Equation (Equation 3), within each one of the nine contexts, the products of these elementary observables is −1. Adding together all nine products of the nine contexts yields the algebraically maximal sum −1 for all quantum value assignments. This is in contradiction to the classical predictions, which never yield −8 or −9. Note that this argument requires the counterfactual existence of all quantum observables ${𝗙}_{i}=|a_{i}\rangle \langle a_{i}|$, even as only a single one context (from nine contexts $C_{1},\ldots ,C_{9}$) is operationally accessible.

## 2. Generalized Operator-Valued Arguments for Mixed States

From now on, we shall assume that states are prepared (preselected) to be in a “maximal” mixture $\rho =\frac{1}{n}{𝟙}_{n}$, where *n* stands for the dimension of the Hilbert space. That is, we abandon state independence for “maximal ignorance” or “maximally scrambled (pure) states”. This cannot be performed from a pure state by merely unitary, one-to-one, means. One has to allow many-to-one processes such as (partial) tracing over constituents of a multipartite state. The advantage of such states is that the expectation value of an operator 𝗔 reduces to the weighted sum over its eigenvalues $\lambda_{1},\ldots ,\lambda_{n}$, that is ${\langle 𝗔\rangle }_{\rho }=\mathrm{Tr}\left(𝗔\rho \right)=\frac{1}{n}\mathrm{Tr}\left(𝗔{𝟙}_{n}\right)=\frac{1}{n}\left(\lambda_{1}+\ldots +\lambda_{n}\right)$.

Then, from a purely algebraic point of view, Householder transformations can be characterized in terms of commutativity ([18], §79, 84): the two observables associated with a pure state and the corresponding expectation values are just functional variations of one and the same maximal operator ([19], Satz 8) (see also [13], Section 4). For an illustration, consider two operators 𝗣 and 𝗘 whose respective eigensystems include identical projection operators, but different eigenvalues.

To be more precise, according to the spectral theorem, let $C=\left\{e_{1},e_{2},\ldots ,e_{n}\right\}\equiv \{|e_{1}\rangle ,|e_{2}\rangle ,\ldots ,|e_{n}\rangle \}$ with n≥2 be an orthonormal basis suitable for a spectral decomposition of 𝗣 and 𝗘, and let ${𝗙}_{i}=|e_{1}\rangle \langle e_{1}|$ be the associated one-dimensional orthogonal projection operators that are mutually orthogonal. Then, the spectral sums of 𝗣 and 𝗘 can be uniformly written as:(4)$$\begin{array}{c} 𝗣=\sum_{i=1}^{n}\lambda_{i}{𝗙}_{i}=\left(+1\right)\cdot {𝗙}_{1}+\left(0\right)\cdot \underset{{𝗙}_{\left\{2,\ldots ,n\right\}}}{\underbrace{\left(\sum_{i=2}^{n}{𝗙}_{i}\right)}}={𝗙}_{1},\\ 𝗘=\sum_{i=1}^{n}\mu_{i}{𝗙}_{i}=\left(-1\right)\cdot {𝗙}_{1}+\left(1\right)\cdot \underset{{𝗙}_{\left\{2,\ldots ,n\right\}}}{\underbrace{\left(\sum_{i=2}^{n}{𝗙}_{i}\right)}}=-{𝗙}_{1}+{𝗙}_{\left\{2,\ldots ,n\right\}}. \end{array}$$
From this perspective of the spectral decompositions, a transition from 𝗣 to 𝗘 is nothing more than a mapping of the eigenvalues in the spectral sums of (Equation 4):(5)$$\begin{array}{c} \left\{\lambda_{1},\lambda_{2},\ldots ,\lambda_{n}\right\}=\left\{1,\underset{n-1\;\mathrm{times}}{\underbrace{0,\ldots ,0}}\right\}\mapsto \left\{\mu_{1},\mu_{2},\ldots ,\mu_{n}\right\}=\left\{-1,\underset{n-1\;\mathrm{times}}{\underbrace{1,\ldots ,1}}\right\}. \end{array}$$
From this spectral point of view, a generalization to mutually disjoint eigenvalues, for instance different primes $p_{1},\ldots ,p_{n}$, suggests itself, such that, in the orthonormal basis, also known as the context, $C=\left\{e_{1},e_{2},\ldots ,e_{n}\right\}\equiv \{|e_{1}\rangle ,|e_{2}\rangle ,\ldots ,|e_{n}\rangle \}$ corresponding to mutually perpendicular orthogonal operators ${𝗙}_{1},\ldots ,{𝗙}_{n}$, the operator associated with the maximal observable has just diagonal entries:(6)$$M=\sum_{i}^{n}p_{i}{𝗙}_{i}=\mathrm{diag}\left(\begin{array}{c} p_{1},\ldots ,p_{n} \end{array}\right).$$
This generalization has the advantage that, because all eigenvalues are prime, all combinations and, in particular, its product $\Pi =p_{1}\cdots p_{n}$, have unique prime decompositions. This translates into a unique decomposition into eigenvalues.

The number of eigenvalues in the spectral sum can be compared with the chromatic number of the sphere [20,21,22], as well as of hypergraphs [14,23]. (Hyper)graphswhose chromatic number exceeds the number of vertices per hyperedge (the clique number) have no classical noncontextual truth assignments formalized by two-valued {0,1} states. This strategy to obtain noncontextual classical colorings of orthogonality (hyper)graphs derived from quantum observables fails for those (hyper)graphs whose chromatic number *n* is equal to the dimension of the associated Hilbert space. These cases also yield no state-independent quantum contextuality, because there exist classical noncontextual observables whose *n* colors can be one-to-one mapped (relabeled) into the observable values $p_{1},\ldots ,p_{n}$.

Another possibility is a choice of the eigenvalues −1,−1,1,1 or any permutation thereof, yielding a quantum prediction of the sum of the products equal to 9·(−1·−1·1·1)=9, which is just the negative of Cabello’s prediction [3].

## 3. Generalized Operations

Other methods to derive state-dependent quantum contextuality involving “maximally mixed states” use operations different from multiplication. The most elementary such operation is the summation among all eigenvalues within a given maximal observable or context. The resulting violations can be tested in a similar (counterfactual) manner as for the sums of products.

For the sake of an example, we again use the Kochen–Specker-type configuration introduced by Cabello, Estebaranz, and García-Alcaine [6], depicted in Figure 2. If instead of multiplying the eigenvalues within any such context (yielding −1·1·1·1=−1), these eigenvalues are added, we obtain the context sum −1+1+1+1=2 (This renders an expectation of the context sum divided by four; that is $\frac{1}{2}$). The associated function between operators within a given context $C_{j}$, 1≤j≤9, is addition:(7)$$g\left({𝗙}_{C_{j},1},{𝗙}_{C_{j},2},{𝗙}_{C_{j},3},{𝗙}_{C_{j},4}\right)=-{𝗙}_{C_{j},1}+{𝗙}_{C_{j},2}+{𝗙}_{C_{j},3}+{𝗙}_{C_{j},4}=S_{C_{j}}$$

As there are nine contexts $C_{j}$, 1≤j≤9, the sum over all context sums is 2·9=18, which is not divisible by four. The respective expectation, given a preselected state $\rho =\frac{1}{4}{𝟙}_{4}$, is:(8)$${\langle \sum_{j=1}^{9}S_{C_{j}}\rangle }_{\rho }=\sum_{j=1}^{9}{\langle S_{C_{j}}\rangle }_{\rho }=\sum_{j=1}^{9}\mathrm{Tr}\left(S_{C_{j}}\rho \right)=\frac{1}{4}\sum_{j=1}^{9}\mathrm{Tr}\left(S_{C_{j}}{𝟙}_{4}\right)=\sum_{j=1}^{9}\frac{1}{2}=\frac{9}{2}.$$

A classical computation produces only multiples of four: Since the 18 observables $a_{1},\ldots ,a_{18}$ are bi-connected—that is, every such observable occurs in exactly two contexts—the sum total of all dichotomic observables is:(9)$$2\left(a_{1}+\cdots +a_{18}\right)=n,\;\mathrm{with}\;a_{1},\ldots ,a_{18}\in \left\{-1,1\right\},\;n\in Z,$$
so that −36≤n≤36. Suppose there are *k* positive observables $a_{i}$ and 18−k negative observables $a_{j}$. Therefore, all cases are permutations of the following configuration:(10)$$\underset{k\;\mathrm{positive}\;a_{i}=1}{\underbrace{a_{1}+\cdots +a_{k}}}+\underset{18-k\;\mathrm{negative}\;a_{j}=-1}{\underbrace{a_{k+1}+\cdots +a_{18}}}=k-\left(18-k\right)=2\left(k-9\right)=\frac{n}{2},$$
with k∈N, so that:(11)$$0\leq k=9+\frac{n}{4}\leq 18,\;\mathrm{and}\;n=-36+4k.$$
This results in *n* arithmetically progressing from −36 in steps of four, that is:(12)$$k\in \left\{0,1,\ldots ,18\right\},\;\mathrm{with}\;\mathrm{respective}\;n\in \left\{-36,-32,\ldots ,0,\ldots ,32,36\right\}.$$

In particular, as 18 is not divisible by four, no sum total of 18 can be classically realized by the configuration of Cabello, Estebaranz, and García-Alcaine [6]. Classical expectations from the assumption of equidistribution of the occurrences are obtained by dividing these cases by four.

Indeed, a combinatorial argument shows that there are:(13)#(n(k))=#(−36+4k)=18k=1818−k=18k!(18−k)!
configurations, yielding n=−36+4k, so that the number of occurrences are #(±0)=48620, #(±4)=43758, #(±8)=31824, #(±12)=18564, #(±16)=8568, #(±20)=3060, #(±24)=816, #(±28)=153, #(±32)=18, #(±36)=1. This classical prediction is in contrast with the quantum prediction 18, which always occurs.

## 4. Applications beyond the Quantum Domain

It would certainly be interesting to study analogs of Householder transformations for systems that are not quantized, but exhibit some form of complementary or contextual behavior. To specify such extensions, one would need to commit to or define the meaning of “contextuality”.

There exist synthetic forms of contextuality that are inspired by Bohr [24,25] and Heisenberg [26]. These allow comprehensive applicability by emphasizing the conditionality of phenomena by the impossibility of any sharp distinction of, or separation between, general empirical objects or entities, in conformity with Bohr’s “interaction with the measuring instruments which serve to define the conditions under which the phenomena appear”. More restricted, analytic notions of contextuality can be defined through various probabilistic violations of classical and nonclassical probability distributions, or from the scarcity, or the lack of, classical value assignments [25,26,27,28,29,30,31,32,33,34,35,36,37].

The general tactic is a transition or recasting from a dichotomic regime—such as {0,1} or {−1,+1} measurement outcomes—into multi-valued observables with more than two outcomes. Multiple values of an observable may “compress” arguments considerably: whereas the information gain per measurement is equal for just two outcomes, it is higher for three or more outcomes even in the single-particle regime. This is because it is always possible to “project” multi-valued outcomes to dichotomic observables by partitioning the set of multiple outcomes into two subsets, a technique used by Meyer [21] based on findings by Godsil and Zaks [20]. Thereby, information is lost, as this kind of projection amounts to a many-to-one mapping for “many” greater than one. In the multi-partite regime, multiplication or other operations of two or more nonzero observables may also reduce the entropy when compared to {0,1}-valued observables [11]. This is because of the skewed, unbalanced effect of multiplication x·y of two values x∈{0,1} and y∈{0,1}, as compared to, say, $E_{x}\cdot E_{y}$ of two values $E_{x}\in \left\{-1,1\right\}$ and $E_{y}\in \left\{-1,1\right\}$.

## 5. Summary

We discussed Householder transformations as a means to recast arguments involving probabilities into expectations of dichotomic observables. By generalizing this procedure, we used the spectral decomposition of the Householder transformation; more explicitly, we allowed eigenvalues not restricted to a single occurrence of minus one and all the others plus one. For instance, dichotomy can be modulated by allowing more than one negative eigenvalue. This allows novel generalized operator-valued arguments for contextuality. We also discussed new forms of state-dependent contextuality by variation of the functional manipulation and relation of the operators. In particular, we considered additivity.

As some original forms of expectation- or operator-based arguments such as Greenberger–Horne–Zeilinger [10,11] or Householder-based state-independent contextuality [3], those arguments developed here use complementary and thus counterfactual observables. Likewise, reasonings involving multiplication or addition of products or sums of the observables within single contexts allow violations of admissibility [5], in particular exclusivity and completeness.

Those considerations inspire new ways of generating and observing nonclassical phenomena. This is not necessarily restricted to quantum contextuality. Thereby, generalized Householder transformations could inspire and expand expressibility and yield advantages through the plasticity of the values of the observable outcomes.

## Figures and Tables

**Figure 1 entropy-24-00429-f001:**
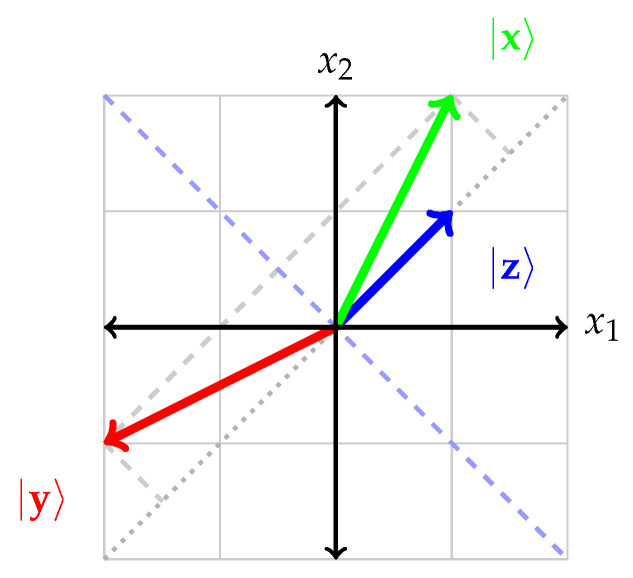
Depiction of the Householder transformation ${𝗨}_{z}$ with $|z\rangle ={\left(\begin{array}{c} 1,1 \end{array}\right)}^{⊺}$ acting on a vector $|x\rangle ={\left(\begin{array}{c} 2,1 \end{array}\right)}^{⊺}$. The resulting “reflected” vector $|y\rangle ={𝗨}_{z}|x\rangle$ and the original vector |x〉 have the same length or norm. Its component along |z〉 is reversed, whereas its component orthogonal to |z〉 remains the same.

**Figure 2 entropy-24-00429-f002:**
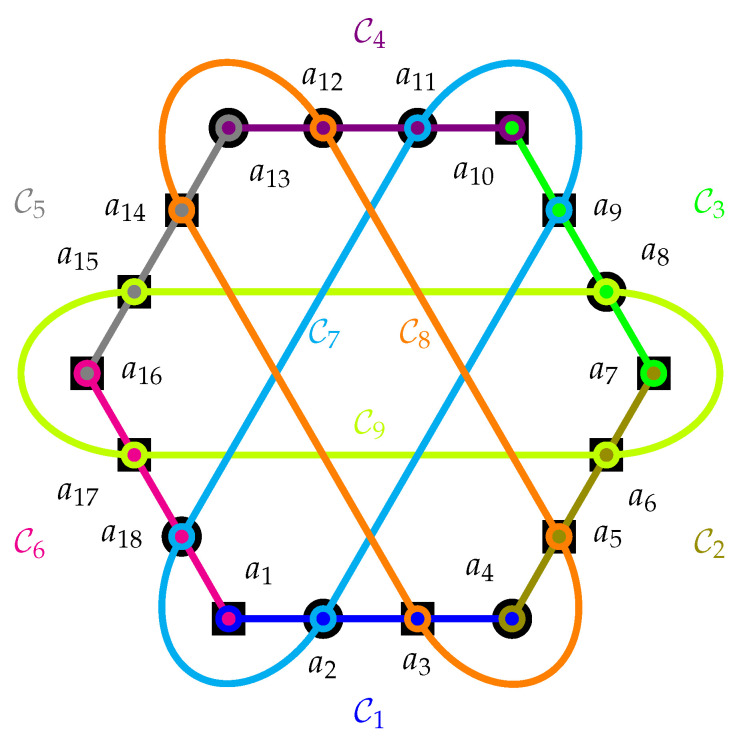
Orthogonality diagram (hypergraph) of a configuration of observables without any two-valued state, used in a parity proof of the Kochen–Specker theorem presented by Cabello, Estebaranz, and García-Alcaine [6]. One (from 9216) underlaid value assignments represents squares as “+1” and circles as “−1”. A quantum realization is, for example, in terms of 18 orthogonal projection operators associated with the one-dimensional subspaces spanned by the vectors from the origin ${\left(0,0,0,0\right)}^{⊺}$ to $|a_{1}\rangle ={\left(\begin{array}{c} 0,0,1,-1 \end{array}\right)}^{⊺}$, $|a_{2}\rangle ={\left(\begin{array}{c} 1,-1,0,0 \end{array}\right)}^{⊺}$, $|a_{3}\rangle ={\left(\begin{array}{c} 1,1,-1,-1 \end{array}\right)}^{⊺}$, $|a_{4}\rangle ={\left(\begin{array}{c} 1,1,1,1 \end{array}\right)}^{⊺}$, $|a_{5}\rangle ={\left(\begin{array}{c} 1,-1,1,-1 \end{array}\right)}^{⊺}$, $|a_{6}\rangle ={\left(\begin{array}{c} 1,0,-1,0 \end{array}\right)}^{⊺}$, $|a_{7}\rangle ={\left(\begin{array}{c} 0,1,0,-1 \end{array}\right)}^{⊺}$, $|a_{8}\rangle ={\left(\begin{array}{c} 1,0,1,0 \end{array}\right)}^{⊺}$, $|a_{9}\rangle ={\left(\begin{array}{c} 1,1,-1,1 \end{array}\right)}^{⊺}$, $|a_{10}\rangle ={\left(\begin{array}{c} -1,1,1,1 \end{array}\right)}^{⊺}$, $|a_{11}\rangle ={\left(\begin{array}{c} 1,1,1,-1 \end{array}\right)}^{⊺}$, $|a_{12}\rangle ={\left(\begin{array}{c} 1,0,0,1 \end{array}\right)}^{⊺}$, $|a_{13}\rangle ={\left(\begin{array}{c} 0,1,-1,0 \end{array}\right)}^{⊺}$, $|a_{14}\rangle ={\left(\begin{array}{c} 0,1,1,0 \end{array}\right)}^{⊺}$, $|a_{15}\rangle ={\left(\begin{array}{c} 0,0,0,1 \end{array}\right)}^{⊺}$, $|a_{16}\rangle ={\left(\begin{array}{c} 1,0,0,0 \end{array}\right)}^{⊺}$, $|a_{17}\rangle ={\left(\begin{array}{c} 0,1,0,0 \end{array}\right)}^{⊺}$, $|a_{18}\rangle ={\left(\begin{array}{c} 0,0,1,1 \end{array}\right)}^{⊺}$, respectively.

## Data Availability

Not applicable.

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
