# Peer review of "Generalized Householder Transformations"

_entropy, 2022, doi:10.3390/e24030429_

Round 1

Reviewer 1 Report

The author discusses some aspects of the Householder transformations. In my opinion, the results presented in the paper are not enough to write a good-quality article in Entropy. Besides, it is hard to tell what is a non-trivial result and what is just a straightforward observation. The structure of the paper requires emphasising the essential results. For example, on page 4, the proof is placed in a regular text. 

Moreover,

  • a typo in line 2 of Section 2
  • page 6, line 9, extra coma before "for instance".
  • the reference section needs a lot of effort to put it in a correct form. 

I cannot recommend the paper for publication.

Author Response

Reply to the report of the 1st Referee (1stR-MDPI _ Reply review report)

I am afraid that I shall not be able to fully satisfy the 1st Referee.

To the criticism that it is unclear what I have achieved in this paper, I would kindly like to refer to the abstract:

"The Householder transformation, allowing a rewrite of probabilities into expectations of dichotomic observables, is generalized in terms of its spectral decomposition. The dichotomy is modulated by allowing more than one negative eigenvalues, or by abandoning binaries altogether, yielding generalized operator-valued arguments for contextuality. We also discuss a form of contextuality by variation of the functional relations of the operators; in particular, by additivity."

I have also added the following paragraph in the introductory section:

"
The respective techniques are well developed but may be less known in the quantum foundations community,
so a review at the beginning seems in order.
We shall then proceed to modifications of Householder transformations to nondichotomic, multiple eigenvalues. "

At first, I was, unfortunately, at a loss for locating "a typo in line 2 of Section 2", as pointed out by the referee. This line reads

"... mally mixed states” use different operations than multiplication. The most elementary"

I have changed the sentence now to

"... mixed states” use operations different from multiplication. ..."

But then I realized that the submission included two pdf files; one of these files exhibited several serious typographic deficiencies; among them was the one mentioned by the Referee.

Thank you for pointing out the extra comma; I have deleted one comma.

I have been again at a loss of understanding what the Referee means when writing "the reference section needs a lot of effort to put it in a correct form."

The reference section has been automatically generated from my bib-file database with an MDPI bst-file format file; I am, of course, happy to correct any deficiency in the database; also the format can be changed easily if advised how, but I am afraid that this would no longer conform with MDPI recommendations.

Only later did I realize that in the "deficient pdf version" of the manuscript, the references started with Reference 24. This should not occur in the actual version.

Maybe some of the uneasiness of the Referee with this manuscript is also due to the many misstated formulae in the pdf version in which the symbol for the unit operator was absent?

I sincerely apologize for the dysfunctional form of the manuscript!

Reviewer 2 Report

This paper expertly dissects and discusses a domain of physics whose understanding is fundmental to progress. It is to be welcomed, and my minor concern should be correctable. It is that the movement from quantum to non-quantum contexuality is not clearly defined. Accordingly, the potential applications of the author's principles to the non-quantum domain are left without adequate discussion. (This application is of sufficient interest to me that I would like to explore it with the author.)

Author Response

Reply to the report of the 2nd Referee (2ndR-MDPI _ Reply review report)

I would like to thank the Referee for a suggestion by stating "... that the movement from quantum to non-quantum contexuality is not clearly defined. Accordingly, the potential applications of the author's principles to the non-quantum domain are left without adequate discussion."

Correspondingly, I have added a section that mentions this possibility:

"\section{Applications beyond the quantum domain}

It would certainly be interesting to study analogs of Householder transformations for systems that are not quantized
but exhibit some form of complementary or contextual behavior.
To specify such extensions, one would need to commit to or define a meaning of ``contextuality''.

There exist synthetic forms of contextuality that are inspired by Bohr~\cite{bohr-1949,Khrennikov2017}
and Heisenberg~\cite{Jaeger2019}. These allow comprehensive applicability by emphasizing the
conditionality of phenomena by the impossibility of any sharp distinction of, or separation between, general empirical
objects or entities; in conformity with Bohr's ``interaction with the measuring instruments which serve to define the conditions
under which the phenomena appear''.
More restricted, analytic notions of contextuality can be defined through various probabilistic violations of classical and nonclassical
probability distributions, or from the scarcity, or the lack of, classical value
assignments~\cite{peres222,svozil-2011-enough,Dzhafarov-2017,Abramsky2018,Grangier_2002,Khrennikov2017,Jaeger2019,Jaeger2020,Auffeves-Grangier-2018,Auffves2020,Grangier-2020,cabello2021contextuality,svozil-2021-context}.

The general tactic is a transition or recasting from a dichotomic
regime---like $\{0,1\}$ or $\{-1,+1\}$ measurement outcomes---into multi-valued observables with more than two outcomes.
Multiple values of an observable may ``compress'' arguments considerably: whereas the information gain per measurement is equal for just two outcomes,
it is higher for three or more outcomes even in the single-particle regime.
This is because it is always possible to ``project'' multiple-valued outcomes to dichotomic observables by partitioning the set
of multiple outcomes into two subsets, a technique used by Meyer~\cite{meyer:99} based on findings by
Godsil and Zaks~\cite{godsil-zaks}. Thereby information is lost, as this kind of projection amounts to a many-to-one mapping for ``many'' greater than one.
In the multi-partite regime,
multiplication or other operations of two or more nonzero observables may also reduce the entropy when compared to $\{0,1\}$-valued observables~\cite{svozil-2020-ghz}.
This is because of the skewed, unbalanced effect of multiplication $x \cdot y$ of two values $x\in \{0,1\}$ and  $y\in \{0,1\}$,
as compared to, say, $E_x \cdot E_y$  of two values $E_x\in \{-1,1\}$ and  $E_y\in \{-1,1\}$."

I am aware that this might be just the beginning of a subject that needs further study, but I hope that this at least partly covers the Referee's suggestions.

I sincerely thank the Reviewer for the suggestion, and for the attention and care dedicated to the manuscript.

Reviewer 3 Report

The paper is a stimulating exploration of using the spectral decomposition of the Householder transformation in order to map probabilities into expectations of dichotomic observables. I do recommend the paper for publication after fixing a few presentation glitches, as indicated in the attached annotated manuscript. The most important one is that the symbol for the identity matrix I does not appear in quite a number of formulas. Other minor misspellings or formatting issues are also indicated in the attachment. Finally, the References start at number 24.

Author Response

Reply to the report of the 3rd Referee (3rdR-MDPI _ Reply review report)

I thank the Referee for the careful reading of the manuscript and the corrections suggested, which I all implemented.

I have to admit that I first was a little puzzled about some of the mistakes mentioned until I realized that, for some reason, a "deficient pdf version" of the manuscript was sent to the Referee. In this version, for instance, the references started with Reference 24. This should not occur in the corrected, actual version.

Many errors originated in the "blank" representation of the identity matrix. I suppose that came about because of my use of "\Eins"  for the identity matrix, with "\Eins" defined in the preamble by a rather lengthy declaration. I have now commented most of this definition, and just use the declaration "\DeclareRobustCommand{\Eins}{\text{\usefont{U}{bbold}{m}{n}1}}" for "\Eins".

As a result, all "\Eins"---standing for the unit matrix---did not appear at all in the deficient version of the manuscript! The Referee clearly pointed this out, and carefully corrected all (wrong) formulae that should contain an identity matrix but did not in the deficient pdf file version.

I am sorry for this problem, and I hope that it does not appear in the revised version.

I very kindly thank the Referee for all the efforts and attention dedicated to the manuscript.